# Detection of SARS-CoV-2 on Surfaces in Households of Persons with COVID-19

**DOI:** 10.3390/ijerph18158184

**Published:** 2021-08-02

**Authors:** Perrine Marcenac, Geun Woo Park, Lindsey M. Duca, Nathaniel M. Lewis, Elizabeth A. Dietrich, Leslie Barclay, Azaibi Tamin, Jennifer L. Harcourt, Natalie J. Thornburg, Jared Rispens, Almea Matanock, Tair Kiphibane, Kimberly Christensen, Lucia C. Pawloski, Alicia M. Fry, Aron J. Hall, Jacqueline E. Tate, Jan Vinjé, Hannah L. Kirking, Eric Pevzner

**Affiliations:** 1U.S. Centers for Disease Control and Prevention, 1600 Clifton Rd., Atlanta, GA 30329, USA; fyt8@cdc.gov (G.W.P.); pgz5@cdc.gov (L.M.D.); pha6@cdc.gov (N.M.L.); wul2@cdc.gov (E.A.D.); gvm3@cdc.gov (L.B.); axt4@cdc.gov (A.T.); zaq6@cdc.gov (J.L.H.); nax3@cdc.gov (N.J.T.); jared.r.rispens@uscg.mil (J.R.); xdf2@cdc.gov (A.M.); ecz6@cdc.gov (L.C.P.); agf1@cdc.gov (A.M.F.); esg3@cdc.gov (A.J.H.); jqt8@cdc.gov (J.E.T.); ahx8@cdc.gov (J.V.); hrj7@cdc.gov (H.L.K.); ecp9@cdc.gov (E.P.); 2Epidemic Intelligence Service, U.S. Centers for Disease Control and Prevention, 1600 Clifton Rd., Atlanta, GA 30329, USA; 3Utah Department of Health, 288 North 1460 West, Salt Lake City, UT 84116, USA; kchriste@utah.gov; 4Salt Lake County Health Department, 610 South 200 East, Salt Lake City, UT 84111, USA; mkiphibane@slco.org

**Keywords:** SARS-CoV-2, fomite transmission, household transmission, prevention

## Abstract

SARS-CoV-2 transmission from contaminated surfaces, or fomites, has been a concern during the COVID-19 pandemic. Households have been important sites of transmission throughout the COVID-19 pandemic, but there is limited information on SARS-CoV-2 contamination of surfaces in these settings. We describe environmental detection of SARS-CoV-2 in households of persons with COVID-19 to better characterize the potential risks of fomite transmission. Ten households with ≥1 person with laboratory-confirmed COVID-19 and with ≥2 members total were enrolled in Utah, U.S.A. Nasopharyngeal and anterior nasal swabs were collected from members and tested for the presence of SARS-CoV-2 by RT-PCR. Fifteen surfaces were sampled in each household and tested for presence and viability of SARS-CoV-2. SARS-CoV-2 RNA was detected in 23 (15%) of 150 environmental swab samples, most frequently on nightstands (4/6; 67%), pillows (4/23; 17%), and light switches (3/21; 14%). Viable SARS-CoV-2 was cultured from one sample. All households with SARS-CoV-2-positive surfaces had ≥1 person who first tested positive for SARS-CoV-2 ≤ 6 days prior to environmental sampling. SARS-CoV-2 surface contamination occurred early in the course of infection when respiratory transmission is most likely, notably on surfaces in close, prolonged contact with persons with COVID-19. While fomite transmission might be possible, risk is low.

## 1. Introduction

COVID-19 is caused by severe acute respiratory syndrome coronavirus 2 (SARS-CoV-2). SARS-CoV-2 transmission occurs primarily through respiratory routes, but transmission from contaminated surfaces, or fomites, has been a concern. SARS-CoV-2 RNA has been detected on surfaces in healthcare [1,2,3,4,5,6,7,8] and community [9,10] settings, residences [11], quarantine rooms [3,12], and a cruise ship [13]; however, attempts to culture virus from environmental samples testing positive for SARS-CoV-2 have not been successful [2,3,6,7,13]. Only studies examining the stability of SARS-CoV-2 on surfaces in controlled laboratory settings have found that viable virus can be recovered from plastic and stainless steel for 3–4 days [14,15] and in one study up to 28 days [16], while persistence on porous surfaces, including cardboard and cotton, is shorter [15,16].

Households have been important sites of SARS-CoV-2 transmission throughout the pandemic [17]. There is a high risk of SARS-CoV-2 transmission from persons sick with COVID-19 to their household members due to prolonged contact [18], but there are limited data on SARS-CoV-2 environmental contamination in these settings. Nested within a study on household transmission of SARS-CoV-2, we conducted environmental sampling in households in Utah during March–April 2020. The goals were to better characterize the presence and viability of SARS-CoV-2 on surfaces in shared living settings and to examine secondary transmission to contacts in households with and without environmental detection of SARS-CoV-2.

## 2. Materials and Methods

### 2.1. Household Identification and Enrollment

This study was conducted as part of a larger study examining secondary household transmission of SARS-CoV-2 [19]. We selected households by convenience sampling in Salt Lake and Davis counties, Utah, during 30 March–25 April 2020, following implementation of stay-at-home orders in those counties. To be eligible, the index patient in a household had to live with ≥1 person, not be hospitalized, and test positive for SARS-CoV-2 by real-time reverse transcription polymerase chain reaction (RT-PCR) on a nasopharyngeal (NP) swab collected ≤10 days prior to enrollment. Households had an initial enrollment visit (day 0), a close-out visit two weeks later (day 14), and in some cases an interim visit or post-close-out visit if a previously SARS-CoV-2-negative household contact developed new symptoms. At each visit, NP and anterior nasal swabs were collected from household members; venipuncture blood specimens were collected on day 0 and day 14. In each household we sampled 15 surfaces (≤700 cm^2^) using Puritan EnviroMax 6” Dry Swabs pre-moistened with phosphate-buffered saline (PBS), as described for norovirus [20]. The protocol and materials for environmental sampling were not available at the beginning of the household investigation. Environmental sampling began as soon as the protocol was approved and the field team received sampling kits. Eight samples were taken from pre-assigned locations: light switches (2 samples/household), toilet handles (1/household), bathroom sink handles (1/household), pillows or nightstands of index cases (1/household), pillows or nightstands of contacts (2/household), and refrigerator handles (1/household). Remaining surfaces were selected by requesting that household members identify frequently touched surfaces in their respective household, which at times were additional surfaces of pre-assigned types (e.g., bathroom sink handles). Questionnaires were administered to collect information on symptoms, exposures, prior SARS-CoV-2 test results, and household prevention measures. Household members logged their symptoms during day 0–day 14. Respiratory symptoms were defined as having ≥1 of runny nose, nasal congestion, sore throat, cough, chest pain, discomfort while breathing, or shortness of breath. Other recorded COVID-19 symptoms were fever (≥38 °C), subjective fever, chills, fatigue, headache, muscle aches, loss of taste or smell, nausea/vomiting, and diarrhea.

### 2.2. Laboratory Testing

NP and nasal swabs were tested for the presence of SARS-CoV-2 RNA by RT-PCR [21]. Serum from blood samples underwent SARS-CoV-2 antibody testing [22]. Environmental swabs were collected and kept at 4 °C during the household visit for ≤4 h until they were transferred to the laboratory and stored at −70 °C in 5 mL PBS. All samples were then shipped to the CDC on dry ice. After the swabs were thawed, vortexed, and briefly centrifuged, 1 mL of eluate was mixed with 4 mL of AVL buffer (Qiagen, Germantown, MD, USA) and 5 µL of A549 cell suspension (10^6^ cells/mL) (ATCC, Manassas, VA, USA) which were added as an extraction control. The remaining 4 mL of eluate were archived at −70 °C for infectivity testing of RT-PCR-positive eluates. Nucleic acid was purified using Midi columns (Omega Biotek, Norcross, GA, USA), concentrated to 50 µL with RNA Clean & Concentrator columns (Zymo Research Corporation, Irvine, CA, USA) [20], and tested by RT-PCR [21]. The archived aliquots of positive swab samples were further tested by cell culture on Vero cells (ATCC CCL-81) to check the infectivity of SARS-CoV-2, with minor modifications from the original protocol [23]. Briefly, swab eluates were filtered (0.45 µm filter, Fisher Scientific, Waltham, MA, USA) and cultured in 96-well plates (200 µL) and T-25 cm^2^ flasks (1 mL) in a humidified 37 °C incubator at 5% CO_2_ for one week, with daily monitoring of virus-induced cytopathic effects (CPE). CPE-positive samples were confirmed for SARS-CoV-2 by RT-PCR [21]. Positive and negative controls were included in all RT-PCR and culture assays as previously described [21,23].

### 2.3. Statistical Analysis

We described frequencies of detecting SARS-CoV-2 on different household surfaces. We also used point-biserial correlation to assess the relationship between environmental detection of SARS-CoV-2 in a household and days elapsed between detection of the most recent SARS-CoV-2 infection in a household member and environmental sampling. Analyses were performed using R (version 4.0.3) [24].

### 2.4. Ethics Statement

This activity was reviewed by the U.S. Centers for Disease Control and Prevention (US CDC) and conducted consistent with applicable federal law and US CDC policy (see, e.g., 45 C.F.R. part 46, 21 C.F.R. part 56; 42 U.S.C. §241(d); 5 U.S.C. §552a; 44 U.S.C. §3501 et seq). Written consent was collected at enrollment. Parental consent was obtained for children < 18 years; assent was obtained from children 7–17 years old.

## 3. Results

We conducted environmental sampling in ten households a median 11 days [range: 2–22] after an index case first tested positive for SARS-CoV-2 (Figure 1). Four households (HH) (40%; HH-07, HH-08, HH-09, HH-10) underwent environmental sampling at the enrollment visit (day 0); five households (50%; HH-02, HH-03, HH-04, HH-05, HH-06) underwent sampling at an interim visit occurring a median 7 days [range: 4–9] after the enrollment visit; and one household (10%; HH-01) underwent sampling on day 20 following the close-out visit. Six (60%) of these ten households had detectable SARS-CoV-2 RNA on ≥1 surface, with a median of two RT-PCR-positive surfaces per household [range: 1–13] (Table 1).

We detected SARS-CoV-2 RNA on 23 (15%) of 150 sampled surfaces, especially on nightstands (4/6 samples, 67%), pillows (4/23, 17%), and light switches (3/21, 14%) (Table 2). All six households with SARS-CoV-2 contamination in the household environment had ≥1 nightstand or pillow with detectable RNA (Table 1). RNA was also detected on high-touch surfaces throughout households: doorknobs (2/17, 12%); kitchen surfaces and appliances, including a sink handle (1/5, 20%), countertop (1/9, 11%), table (1/4, 25%), refrigerator handle (1/11, 9%), microwave (1/7, 14%), and trash can lid (1/1, 100%); and electronic items, including a phone (1/3, 33%), computer (1/4, 25%), and TV remote control (1/7, 14%). We found SARS-CoV-2 RNA infrequently in bathrooms; none of the 13 toilet handles and 1 (8%) of 13 bathroom sink handles sampled had detectable RNA. The median cycle threshold (C_t_) value of SARS-CoV-2-positive surfaces in households was 33.8 [range: 26.4–37.2]. While most SARS-CoV-2-positive surfaces were made of non-porous materials (metal, plastic, and treated wood), cloth pillows also had detectable RNA with a median C_t_ value of 35.6 [range: 32.8–36.4].

We used all 23 RT-PCR-positive environmental samples to inoculate Vero cells; in one (4%) of 23 samples, we observed CPE and confirmed SARS-CoV-2 recovery. This sample came from a nightstand swab (C_t_ = 26.4) belonging to an index case (Case ID 09-00, a 35-year-old man) with respiratory symptoms whose NP swab was positive for SARS-CoV-2 (C_t_ = 15.5) on the environmental sampling date. This man first tested positive for SARS-CoV-2 two days prior to environmental sampling and lived in a household with a 100% secondary attack rate (HH-09, Figure 1). Two household members (Case IDs 09-01, 09-02) tested positive for SARS-CoV-2 for the first time on the day of environmental sampling, and 13 (87%) of 15 sampled surfaces in this household were positive for SARS-CoV-2 by RT-PCR (Table 1). No other swabs yielded recovery of live virus.

We assessed differences in characteristics and timing of human cases, as well as transmission dynamics and prevention measures, between households with and without the detection of SARS-CoV-2 RNA on surfaces. Among the six households with detectable SARS-CoV-2 RNA in the environment, the median number of days elapsed between the first RT-PCR-positive test in the most recent COVID-19 case and the day of environmental sampling was 4 days [range: 0–6] (Figure 1). Six (86%) of the seven most recent COVID-19 cases in these six households (Case IDs 06-01, 07-00, 08-00, 09-01, 09-02, 10-00) were positive on the day of environmental sampling with a median C_t_ value of 27.7 [range: 22.6–36.5]. Four (66%) of these six SARS-CoV-2-positive persons were experiencing symptoms at the time of environmental sampling, while two (33%) were pre-symptomatic (Case IDs 09-01, 09-02). Among the four households with no environmental detection of SARS-CoV-2, the median days elapsed between the first RT-PCR-positive test in the most recent COVID-19 case and the day of environmental sampling was 12 days [range: 11–12]. Two household members (Case IDs 04-00 and 05-00) in two (50%) of these four households tested positive on the day of environmental sampling with a median C_t_ value of 30.9 [range: 30.3–31.5], and both (100%) were experiencing symptoms at the time of environmental sampling. Overall, environmental SARS-CoV-2 RNA detection was negatively correlated with the number of days between the first SARS-CoV-2-positive RT-PCR test of the most recently infected household member and environmental sampling (point-biserial correlation r = −0.94, *p* < 0.001).

Of six households with environmental detection of SARS-CoV-2, three (50%) had ≥1 instance of secondary transmission from the index COVID-19 case to a household contact as detected via RT-PCR on NP and nasal swabs and confirmed by serology: HH-01, HH-06, and HH-09 (Figure 1). All three (100%) of these households were unable to use isolation measures for sick persons, and none reported routine cleaning or disinfection of common areas. Of the three households with detectable SARS-CoV-2 RNA in the environment but no secondary transmission, two (67%) reported taking isolation measures and two (67%) reported using disinfecting wipes and sprays on high-touch surfaces. We found no cases of secondary transmission in the four households that did not have detectable SARS-CoV-2 RNA in the environment. These four households reported that infected persons isolated themselves from other household members using ≥1 of the following strategies: sleeping in separate bedrooms (4/4, 100%), using separate bathrooms (4/4, 100%), or eating separately from household members (3/4, 75%). Two (50%) of these four households reported using disinfecting wipes and sprays on high-touch surfaces after someone became ill with COVID-19.

## 4. Discussion

We detected SARS-CoV-2 RNA most frequently on nightstands and pillows used by persons who recently tested positive for SARS-CoV-2. We also found viral RNA on high-touch surfaces, including light switches, doorknobs, and kitchen appliances. We observed CPE and recovered viable SARS-CoV-2 from the nightstand of a symptomatic man with a high viral load (NP C_t_ = 15.5) and who was early in his course of illness. To our knowledge, this is the first recorded instance of viable SARS-CoV-2 recovery from an environmental swab in a real-life, non-laboratory setting.

Environmental detection of SARS-CoV-2 RNA in a household was strongly correlated with fewer days since the most recent COVID-19 diagnosis in that household. Both households with and without environmental SARS-CoV-2 contamination had symptomatic persons testing positive for SARS-CoV-2 on the day of environmental sampling. However, only households with persons who first tested positive ≤ 6 days prior to environmental sampling had detectable SARS-CoV-2 in the household environment, with one of these households having pre-symptomatic persons. Indeed, a previous study showed no difference in the rate of SARS-CoV-2 contaminated surfaces in rooms occupied by symptomatic and asymptomatic persons at the time of environmental sampling [13]. Our data are consistent with studies showing that viral shedding from persons infected with SARS-CoV-2 peaks early in the course of infection [25,26] and that environmental contamination with SARS-CoV-2 is more likely when patient viral loads are at their highest [1]. Taken together, our data suggest that surface contamination with SARS-CoV-2 occurred through direct contamination with respiratory fluids following close, prolonged contact with sick persons early in their course of illness, particularly in the case of pillows and nightstands. Contamination may also have occurred indirectly when sick persons touched these surfaces after contaminating their hands with their respiratory fluids. Our results also suggest that temporal proximity to when persons first tested positive for SARS-CoV-2 may be a predictor for environmental contamination in households.

Among the six households with environmental detection of SARS-CoV-2, three had secondary transmission from the index case to another household member, including one household with recovery of viable virus. The four households with no environmental SARS-CoV-2 detection had no instances of secondary transmission. The three households with secondary transmission were not able to take isolation precautions for sick persons. It is therefore challenging to determine whether SARS-CoV-2-positive surfaces were indicative of the high viral burden of persons with SARS-CoV-2 in these households that resulted in environmental contamination without contributing to transmission, or if fomite transmission could have potentially contributed to secondary transmission in these households. However, combined with previous studies showing no recovery of viable virus from environmental swabs in healthcare settings [2,3,6,7], quarantine rooms [3], and cruise ship rooms [13], our recovery of viable SARS-CoV-2 on only one (4%) of 23 PCR-positive surfaces supports that the risk of fomite transmission is low. Secondary transmission of SARS-CoV-2 in households likely occurred through respiratory transmission rather than fomite transmission.

Our study is subject to several limitations, including the small sample size of households and differential timing of environmental sampling relative to when index cases first tested positive for SARS-CoV-2 across households. In addition, secondary cases could have occurred from exposures outside the household, though we documented no known community exposures among household contacts who became positive during this study, and stay-at-home orders in Utah reduced this likelihood. Finally, we enrolled households using convenience sampling, so results may not be generalizable to all households with persons with COVID-19.

Our results highlight the importance of existing guidance for persons who may be sick with COVID-19 to wear a mask and isolate from others in their household as soon as possible by staying in a separate room, using a separate bathroom, and avoiding contact with others if possible [27]. All household members should wash their hands often and clean and disinfect high-touch surfaces and items in close proximity to sick persons daily [28]. These measures not only reduce the likelihood of respiratory transmission but might also minimize contamination of surfaces with SARS-CoV-2.

## 5. Conclusions

SARS-CoV-2 is detectable in the household environment of persons with COVID-19, notably on surfaces in close, prolonged contact with persons who recently tested positive for SARS-CoV-2. Our results highlight that viable virus can be recovered from surfaces in natural, non-laboratory settings. Based on our data, the risk of fomite transmission is low. More research is needed to fully evaluate the risk of fomite transmission, especially in the context of widespread vaccination efforts and the spread of new SARS-CoV-2 variants. However, taking measures to reduce respiratory transmission of SARS-CoV-2, including isolation of sick persons, frequent handwashing, and frequent disinfection and cleaning of common surfaces, could minimize surface contamination with SARS-CoV-2 and further reduce the possibility of fomite transmission.

## Figures and Tables

**Figure 1 ijerph-18-08184-f001:**
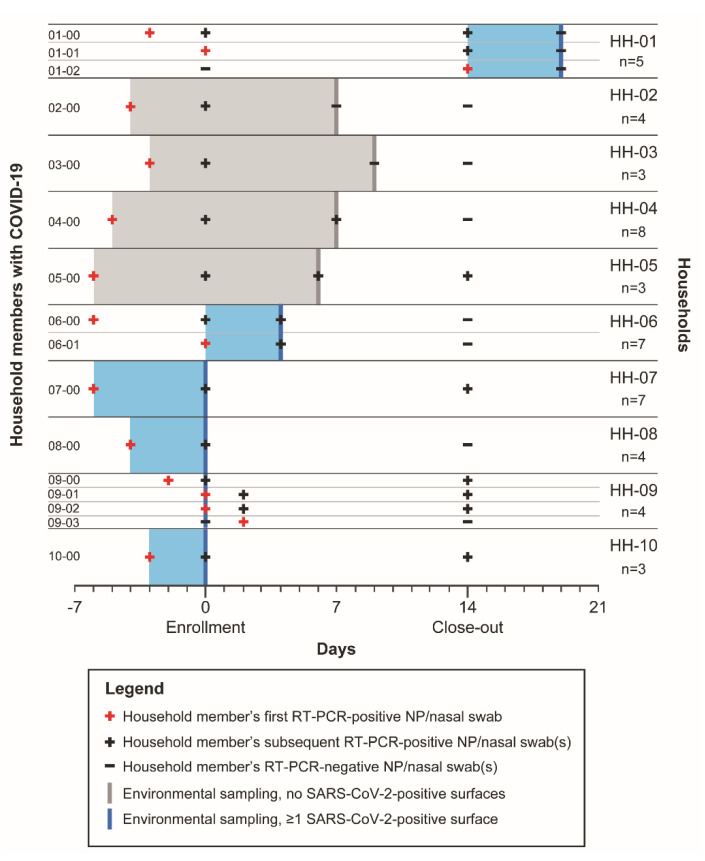
Timing of RT-PCR-positive SARS-CoV-2 tests among household (HH) members and environmental detection of SARS-CoV-2 in households. SARS-CoV-2 RT-PCR-positive tests from nasopharyngeal (NP) or nasal swabs for each household member with COVID-19 (left) in each household (right) are designated with “plus” symbols, with red symbols designating each person’s first SARS-CoV-2-positive test, and black “plus” symbols designating subsequent positive tests. Negative tests are designated with black “minus” symbols. Only household members who tested positive for SARS-CoV-2 over the course of the study are graphed, and “n” designates the total number of members per household inclusive of persons who never tested positive during the study. Secondary transmission is observable in households with contacts testing positive for SARS-CoV-2 by RT-PCR on their NP or nasal swabs for the duration of the study. Environmental sampling days are designated with thick vertical lines, and days elapsed between the first positive test in the most recent household member with COVID-19 and environmental sampling is designated with shaded bars. Blue indicates that SARS-CoV-2 RNA was detected in the household environment, while gray indicates that no surfaces tested positive for SARS-CoV-2 RNA. Note: Shaded bars precede enrollment (day 0) if the most recent household member with COVID-19 first tested positive prior to enrollment. HH-09 has a thick vertical blue line but does not have a shaded bar as the most recent household members with COVID-19 (Case IDs 09-01 and 09-02) tested positive for the first time on the day of environmental sampling (day 0). No NP or nasal swab was collected from the index case in HH-09 (Case ID 09-00) on day 2.

**Table 1 ijerph-18-08184-t001:** Location and cycle threshold (Ct) ^a^ value of SARS-CoV-2 detected at environmental surfaces sampled in each of ten households (HHs) with ≥1 laboratory-confirmed case of COVID-19.

Household ID	Number (% ^b^) of Sampled Surfaces with Detectable SARS-CoV-2 RNA	HH Surfaces with Detectable SARS-CoV-2 RNA (C_t_)
HH-01	2 (13)	Pillow of secondary HH case (36.4); phone (37.0)
HH-02	0 (0)	..
HH-03	0 (0)	..
HH-04	0 (0)	..
HH-05	0 (0)	..
HH-06	3 (20)	Light switch (37.2); pillow of index case (35.0); trash can lid (32.1)
HH-07	2 (13)	Nightstand of index case (34.1); computer (36.6)
HH-08	1 (7)	Pillow of other HH member (36.2)
HH-09	13 (87)	2 light switches (29.6, 33.4); refrigerator handle (29.3); nightstand of index case (26.4), nightstands of 2 secondary HH cases (33.8, 35.7); 2 doorknobs (29.8, 30.6); kitchen counter (33.2); microwave handle (31.8); kitchen sink handle (34.8); furniture (34.7); TV remote control (28.8)
HH-10	2 (13)	Pillow of index case (32.8); bathroom sink handle (34.8)

^a^ Ct value is reported as the mean of Ct values from 2 SARS-CoV-2 genes (N1 and N2). High Ct values indicate there is less viral RNA, while low Ct values indicate more viral RNA. ^b^ Fifteen surfaces were sampled in each household.

**Table 2 ijerph-18-08184-t002:** Location and median cycle threshold (C_t_) ^a^ value of SARS-CoV-2 detected at the environmental surfaces sampled across ten households (HHs) with ≥1 laboratory-confirmed case of COVID-19.

Environmental Surface	Number of Surfaces Sampled ^b^	Number of Surfaces Positive for SARS-CoV-2 (%)	Number of HHs Sampled for Surface (%)	Number of Sampled HHs with SARS-CoV-2-Positive Surface (%)	Median C_t_ [Range]
**Light switch**	21	3 (14)	10 (100)	2 (20)	33.4 [29.6–37.2]
**Refrigerator handle**	11	1 (9)	10 (100)	1 (10)	29.3
**Toilet handle in bathroom**	13	0 (0)	9 (90)	0 (0)	..
**Bathroom sink handle**	13	1 (8)	10 (100)	1 (10)	34.8
**Pillow** ^**c**^ **of:**	23	4 (17)	8 (80)	4 (50)	35.6 [32.8–36.4]
Index case	8	2 (25)	8 (80)	2 (25)	33.9 [32.8–35.0]
Secondary HH case	2	1 (50)	1 (10)	1 (100)	36.4
Other HH member	13	1 (8)	7 (70)	1 (14)	36.2
**Nightstand ^c^ of:**	6	4 (67)	2 (20)	2 (100)	34.0 [26.4–35.7]
Index case	2	2 (100) ^d^	2 (20)	2 (100)	30.2 [26.4–34.1]
Secondary HH case	2	2 (100)	1 (10)	1 (100)	34.8 [33.8–35.7]
Other HH member	2	0 (0)	1 (10)	0 (0)	..
Doorknob/handle	17	2 (12)	10 (100)	1 (10)	30.2 [29.8–30.6]
Kitchen countertop	9	1 (11)	9 (90)	1 (11)	33.2
Microwave handle/button	7	1 (14)	7 (70)	1 (14)	31.8
Kitchen sink handle	5	1 (20)	5 (50)	1 (20)	34.8
Trash can lid	1	1 (100)	1 (10)	1 (100)	32.1
Furniture ^e^	4	1 (25)	4 (40)	1 (25)	34.7
Phone (mobile, landline)	3	1 (33)	3 (30)	1 (33)	37.0
Computer (mouse, keyboard)	4	1 (25)	3 (30)	1 (33)	36.6
TV remote control	7	1 (14)	7 (70)	1 (14)	28.8
Miscellaneous electronics ^f^	4	0 (0)	4 (40)	0 (0)	..
Banister	1	0 (0)	1 (10)	0 (0)	..
Cat litter box	1	0 (0)	1 (10)	0 (0)	..

Note: Locations in bold were pre-assigned. Other surfaces were selected by identifying other high-touch surfaces in each household. ^a^ C_t_ value is reported as the mean of C_t_ values from 2 SARS-CoV-2 genes (N1 and N2). High C_t_ values indicate there is less viral RNA, while low C_t_ values indicate more viral RNA. ^b^ A total of 150 swabs were collected. The rows labeled “Pillow of:” and “Nightstand of:” contain summary data for the total pillows (23) and nightstands (6) swabbed from index cases, secondary HH cases, and other HH members that are also individually listed in the subsection. This column sums to 179 because both summary and individual pillow and nightstand swab data are listed. ^c^ Staff were asked to sample either the pillow or the nightstand of index cases and household members. ^d^ Viable SARS-CoV-2 was recovered from one of these nightstands with C_t_ = 26.4 belonging to 09-00. ^e^ Two tables, a chair, and a baby gate. A kitchen table tested positive. ^f^ A tablet, washing machine button, video game controller, and thermometer.

## Data Availability

The data presented in this study are available upon reasonable request to J.E.T. (jqt8@cdc.gov) and H.L.K. (hrj7@cdc.gov). Requesters will be asked to submit proposals that will undergo review at the US CDC, and all approved data requesters will need to sign a data use agreement.

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
