# Peer review of "Detection of SARS-CoV-2 on Surfaces in Households of Persons with COVID-19"

_ijerph, 2021, doi:10.3390/ijerph18158184_

Round 1
Reviewer 1 Report
Authors submitted a document with a title: Detection of SARS-CoV-2 on surfaces in households of persons 2 with COVID-19.
This document explains the detection in inanimate surfaces inside houses. Their findings are very interesting, and conclusions will help to take preventive actions for health takers/providers when COVID-19 diagnosis is confirmed.
The information about diagnosis timing looks like it is a better predictor for virus environmental presence. This type of information is like infectious bronchitis virus (IBV) which is a coronavirus too.
It could be very interesting if authors could increase sample numbers and if they sampled dishes, glasses and cutlery.
Authors should be more clear about this statement in L247 “the risk of fomite transmission seems to be low.”
Reviewer 2 Report
I agree with the authors that the role of environmental contamination, and in your manuscript more specific fomites, is not well known yet. Therefore I stress the importance of the topic, especially in household settings, in which transmission takes often place. However, I have some minor suggestions for improvement, mostly on clarification of the study design chosen, the presentation of results, and the clarity of some sentences. Below you will find some comments in more detail.
Abstract
Line 25: If I understood correctly, you only included households with more then 1 person, so please state that in the abstract as well.
Introduction
Line 48-51: This sentence is too long and therefore hard to read. Please divide in two shorter sentences.
Materials and Methods
Line 57-60: This sentence is too long and therefore hard to read. Please divide in two shorter sentences.
Line 60-64: This sentence is too long and therefore hard to read. Please divide in two shorter sentences.
Line 64-66: What was the rationale for this non-uniform sampling scheme? What were the criteria for an interim visit?
Line 68-70: Why were environmental samples taken only cross-sectionally? You would have had the opportunity to collect repeated samples. This would have increased your samples size. I also cannot find the reasoning why environmental samples were take on a certain timepoint and why this differs between households. Please explain.
Line 70: Is PBS known to be the best medium for recovery of SARS-CoV-2? What size surface was sampled? How were samples stored and transported directly after sampling. What was the time between sampling and storage at -80? This is all really interesting information, since you have managed to culture one of the samples.
Line 74: Was fever no symptom of interest?
Line 78: Which amount of AVL was used?
Line 81: What was used for culturing. The remainder of AVL, how much fluid was used?
Results
Figure 1: I think you did a good job presenting the results in one figure, I do suggest put some extra information in and try to clarify some aspects.
- I believe NP were collected from household members at each visit? Please put in negative results explicitly by putting in a ‘-’. This will give a better overview and will make clear as well how many household members were included.
- It would be good to include the number of positive environmental samples. In household 9, 13 positive environmental samples were found. This is more than 50% of the total amount of positives. It would be informative to grasp this from the figure.
- Again, the timing for environmental sampling looks completely arbitrary.
- In black and white print the figure cannot be interpreted.
Line 163-168: It is not unlikely that the RNA found is rest materials from the infection and no active shedding anymore. This can be hypothesized in the discussion as reason for no environmental shedding, regardless PCR positive NP.
Discussion
Line 198-202: This sentence is too long and therefore hard to read. Please divide in two shorter sentences.
Line 208-212: This sentence is too long and therefore hard to read. Please divide in two shorter sentences.
Line 212-215: I think you sample size is too small to suggest this. Not enough pre-symptomatic persons were included to get a representative insight.
Line 216-228: Although I agree with you on the statement that the risk of fomite transmission is likely low, I believe you can be a bit more explicit regarding the most likely pathways. It is highly likely that the amount of viral load is responsible for the secondary cases through closely living together (known transmission route through droplets) as well as for the environmental contamination. There is a smaller chance that environmental contamination is an intermediate in the pathway rather than a consequence.
Reviewer 3 Report
This manuscript describes a small study that investigated the detection of SARS-CoV-2 on household surfaces of persons with COVID-19 (as confirmed by RT-PCR). According to the abstract, 10 household were investigated with six households having detectable SARS-CoV-2 RNA on ≥1 surface. In all, SARS-CoV-2 RNA was detected in 23 out of 150 environmental samples and viable SARS-CoV-2 was cultured from one sample. All households with positive surfaces had at least 1 person who tested positive for SARS-CoV-2 less than 6 days before environmental sampling occurred. According to the authors , this suggests that surface contamination occurs early during infection when respiratory transmission is most likely and that risk from fomite transmission is very low.
There are some problems with the description of the methods and results.
- There is no mention of a standardised protocol for environmental sampling. Table 1 indicates that some surfaces were pre-assigned and some were selected at the discretion of the investigating staff. Details of this protocol must be included in the methods as the interpretation of the results depends upon this protocol. The authors need to present their results (and analysis) into two based upon whether the samples were pre-assigned or not. It is not appropriate to include the ad hoc samples in an overall analysis as there is no way of knowing whether the same samples collected in other households would have been positive or not.
- There is no data on positive controls for the environmental sampling and viral culture. How do we know that negative results are not a result of difficulties in sampling or culturing?
- The abstract indicates that 15 surfaces were sampled in each household (ie a total of 150 samples) but the results of 179 samples are presented in Table 1. This needs to be clarified. It is thus unclear how many samples were taken from each household and this is a major confounder for the analysis as presumably it is more likely to detect SARS-CoV-2 RNA the more samples are taken from an individual household.
- Figure 1 is confusing as it seems to imply that environmental sampling was carried out before people were enrolled into the study. How is this possible?
- No data is provided on the number of positive samples per individual household and any analysis of such positive samples (ie were there any patterns in positivity?). This needs to be included
Round 2
Reviewer 3 Report
The authors have clearly addressed the points I raised.